# Differential role of planar cell polarity gene Vangl2 in embryonic and adult mammalian kidneys

Ida Derish[1], Jeremy K. H. Lee[1°], Melanie Wong-King-Cheong[1°], Sima Babayeva[1], Jillian Caplan[1], Vicki Leung[2], Chloe Shahinian[1], Michel Gravel[2], Michael R. Deans[3], Philippe Gros[2], Elena Torban[1]*

**1** Department of Medicine, McGill University and McGill University Health Center Research Institute, Montreal, Quebec, Canada, **2** Department of Biochemistry, McGill University, Montreal, Quebec, Canada, **3** Division of Otolaryngology, Department of Surgery, University of Utah School of Medicine, Salt Lake City, UT, United States of America

☯ These authors contributed equally to this work.
* elena.torban@mcgill.ca

**Data Availability Statement:** All relevant data are within the manuscript and its Supporting Information files.

## Abstract

Planar cell polarity (PCP) pathway is crucial for tissue morphogenesis. Mutations in PCP genes cause multi-organ anomalies including dysplastic kidneys. Defective PCP signaling was postulated to contribute to cystogenesis in polycystic kidney disease. This work was undertaken to elucidate the role of the key PCP gene, Vangl2, in embryonic and postnatal renal tubules and ascertain whether its loss contributes to cyst formation and defective tubular function in mature animals. We generated mice with ubiquitous and collecting duct-restricted excision of Vangl2. We analyzed renal tubules in mutant and control mice at embryonic day E17.5 and postnatal days P1, P7, P30, P90, 6- and 9-month old animals. The collecting duct functions were analyzed in young and adult mutant and control mice. Loss of Vangl2 leads to profound tubular dilatation and microcysts in embryonic kidneys. Mechanistically, these abnormalities are caused by defective convergent extension (larger tubular cross-sectional area) and apical constriction (cuboidal cell shape and a reduction of activated actomyosin at the luminal surface). However, the embryonic tubule defects were rapidly resolved by Vangl2-independent mechanisms after birth. Normal collecting duct architecture and functions were found in young and mature animals. During embryogenesis, Vangl2 controls tubular size via convergent extension and apical constriction. However, rapidly after birth, PCP-dependent control of tubular size is switched to a PCP-independent regulatory mechanism. We conclude that loss of the Vangl2 gene is dispensable for tubular elongation and maintenance postnatally. It does not lead to cyst formation and is unlikely to contribute to polycystic kidney disease.

## Introduction

Congenital anomalies of the kidney and urinary tract (CAKUT) are a subset of birth defects comprising 20–30% of prenatally-identified malformations [1]. Cystic kidney dysplasia is one of the most prevalent CAKUT defects and characterized by microcysts in embryonic kidneys

**Funding:** This study was supported by the grants to E.T. from the Kidney Foundation of Canada (https://kidney.ca) grants # KFOC1518 & KFOC1719 and from the Canadian Institute of Health Research (https://cihr-irsc.gc.ca) grant # CIHR230929. The funders had no role in study design, data collection and analysis, decision to publish, or preparation of the manuscript.

**Competing interests:** The authors have declared that no competing interests exist.

and progressive cyst enlargement during childhood. CAKUT often leads to end-stage renal insufficiency requiring kidney transplantation in adult life [2, 3]. Since CAKUT is heterogeneous, it is important to understand the normal mechanisms that regulate kidney development and cause wide-ranging renal malformations when perturbed.

Kidney development relies on many key developmental pathways [4] including the planar cell polarity (PCP) pathway [5]. PCP was originally discovered in *Drosophila melanogaster* where planar polarity is evident in the uniform orientation of the fly hair on the wing cells [6]. Subsequent studies in *Xenopus laevis* and other vertebrates revealed that the PCP signaling pathway is highly-conserved across vertebrate species [7–10]. In vertebrates, PCP signaling regulates morphogenetic processes such as convergent extension (CE), which organizes cell migration and intercalation. CE is important for organs such as the heart, lungs and kidneys which all have intrinsic tubular structures [11, 12]. Karner *et al* showed that defective CE in the Wnt9b (PCP ligand) knockout mouse causes tubular dilatation and cystic transformation during embryonic kidney development [13]. Over a decade ago, Fischer *et al* described uniform orientation of cell division (OCD) in the growing renal tubules of young mice and rats; OCD was lost in postnatal rodent models of polycystic kidney disease [14]. This discovery paved the way to the postulate that defective PCP signaling randomizes OCD and contributes to renal tubule dilatation and/or cyst initiation [15].

The PCP pathway transduces signals between and within the cells via a set of evolutionarily-conserved proteins [9]. A murine model *Looptail*, *Lp* (a spontaneous S464N mutation in the key PCP gene *Van Gogh-like 2*, *Vangl2*) exemplifies the impact of defective PCP signaling on mouse development: homozygous *Lp* mice exhibit multi-organ defects including a characteristic neural tube defect, craniorachischisis, and intrauterine death by E18.5 [16]. We and others previously reported that embryonic kidneys of homozygous *Lp* mice are misshapen and display hypoplasia of the medullary zone, abnormal ureteric bud (UB) branching, dilated tubules and abnormal glomeruli [17–20]. Branching morphogenesis of the lung and mammary glands were also abnormal in *Lp* embryos [21, 22].

Tubulogenesis and tubule bending are dependent on tightly coordinated apical constriction (AC). The apical surface of each cell shrinks to create a trapezoidal shape that allows cell alignment around the tubular lumen, while the basolateral surface forms the larger outer circumference [23]. AC is required for neural tube bending [24]. In the pronephros of *Xenopus laevis*, trapezoidal cells form cellular rosettes clustered around a potential lumen; cell movement in the perpendicular axis enables tubular elongation while adjusting the apical surface to determine tubular diameter [25]. Ossipova *et al* found that *Vangl2* controls AC of invaginating blastopore lip cells during *Xenopus* gastrulation [26, 27]. Another key PCP gene, Celsr1, was shown to regulate AC in the chick otic placode, as it bends to form the optic vesicle [28].

To assess the requirement for PCP signaling during kidney development, for postnatal tubular modeling and in adult tubular maintenance, we generated mutant mouse strains with ubiquitous and collecting duct-restricted excision of the *Vangl2* gene. We found profound tubular dilatation/cystic phenotype in embryonic kidneys in all mutant strains and established that this is due to abnormal CE and AC. However, kidney-specific loss of *Vangl2* did not alter final tubular size, cell shape, or collecting duct function in adult mice. We propose that the PCP pathway regulates CE and AC during embryonic tubulogenesis, but a developmental switch occurs at the time of birth, allowing for a resolution of the embryonic defect in collecting tubules. Our study establishes differential requirements for *Vangl2* before and after birth. It also refutes the proposed cystogenic role for defective PCP signaling in polycystic kidney disease.

## Materials and methods

### Animal breeding and husbandry

All animal work was conducted according to the Canadian Animal Care Guidance with approval by the Animal Care Committee of McGill University Health Center Research Institute, ACC protocol 7606. *Vangl2* heterozygous mice for the Floxed and Exon4 excision ("Δ") alleles [20] were sibling mated to produce *Vangl2*$^{Δ/Δ}$, *Vangl2*$^{Fl/Fl}$ or *Vangl2*$^{Δ/Fl}$ mice. A morning plug after overnight mating was counted as 0.5 day *postcoitum* (embryonic (E) day 0.5). Conditional *Vangl2* mice with an excision of *Vangl2* in the collecting duct were generated by crossing *Vangl2*$^{Δ/Fl}$ mouse with *HoxB7-Cre-EGFP* deleter mouse [29] (kindly provided by Dr. Carlton Bates, University of Pittsburgh, PA) and then backcrossed to *Vangl2*$^{Δ/Fl}$ to obtain *Hoxb7-Cre+;Vangl2*$^{Δ/CD}$ (thereafter referred to as *Vangl2*$^{Δ/CD}$). *Cre-;Vangl2*$^{Fl/Fl}$ and *Hoxb7-Cre+;Vangl2*$^{+/+}$ mice were used as controls. The *Vangl2*$^{Δ/Fl}$ mice were on the C57/Bl6 background; *Hoxb7-Cre* mice were on the mixed Bl6/CD-1 background; high efficiency of the Hoxb7-Cre allele on the CD1 was previously reported [30]. To produce viable embryos of all genotypes, time-pregnant dams were sacrificed at E17.5 and whole embryos were processed and stored for further analysis. Postnatal *Vangl2*$^{Δ/CD}$ mice and matching controls were sacrificed by cervical dislocation at postnatal (P) day 1, P7, P30, P90, 6 and 9 months, their kidneys were excised and preserved for further analysis. *Looptail*$^{S464N}$ mice, *Lp*, [16] were maintained as heterozygotes and sibling mated to produce E17.5 *Vangl2*$^{Lp/Lp}$ embryos and matching wild-type littermates. *Lp* mice were on the mixed C57BL/6J background. The generation of mice with Pax2-Cre-driven excision of *Vangl2* gene, *Pax2-Cre;Vangl2*$^{ΔTMs/LoxP}$ mice, was previously described [31]: the "ΔTMs" allele is a ubiquitous excision of Exon 4 which encodes four transmembrane domains of Vangl2 protein; the "LoxP" allele allows a tissue-specific excision of Exons 2 and 3 which encode *Vangl2* ATG translation initiation site. Kidneys from the control and conditional *Pax2-Cre;Vangl2*$^{ΔTMs/LoxP}$ mice were provided by Dr. Deans (University of Utah School of Medicine, Salt Lake City, UT); morphological analysis was conducted only on the kidneys isolated from P1 and P15 mice as described below.

### Genotyping

The mouse DNA was isolated from a tail biopsy with the Fast-New Genotyping kit (GT-003, ZmTech Scientifique, Montreal, QC, Canada); the DNA samples were analyzed by polymerase chain reaction (PCR) using the following primers: to detect "floxed" allele 1F primer 5′-TCT TGATTTGTGGCCCAGGCTGAT-3′ and 1R primer 5′-TGCTCAGCCAAGATTGGGAACTCT-3′; to detect excision of Exon 4 Δ allele 2F primer 5′-GACATGTATCACCTCACTTGGCTGGA-3′ and 2R primer 5′-CTTACTATGTGCAAACCACCTTC-3′; to detect Cre allele, Cre-F primer 5′-AGGTTCGTGCACTCATGGA-3′ and Cre-R primer 5′-TCGACCAGTTTAGTTACCC-3′.

The *Vangl2 Lp* mice (S464N allele) were genotyped by visual inspection and Sanger sequencing: the homozygous embryos were identified by the presence of a typical neural tube defect, craniorachischisis; heterozygous mice—by the presence of a characteristic looped tail; the wildtype mice—by the normal appearance. The following primers were used to amplify Exon 8 of murine Vangl2 gene which contains the S464N mutation: mVangl2–Ex8-F 5′-ACC TTAGAAACACCCTAGCT-3′ and mVangl2–Ex8-R 5′-ACAGAGGTCTCCGACTGCAGC-3′, followed by nucleotide sequencing.

### Preparation of the primary collecting duct cells

Kidneys from P5 *Vangl2*$^{Δ/CD}$ and *Cre+;Vangl2*$^{+/+}$ mice were dissected out, cut in half and pelvic/medullary zones were separated from cortices. The separated tissues were enzymatically

dispersed for 2 hours at 37˚C in DMEM/F12 medium (Wisent, Saint-Bruno, QC, Canada), supplemented with 0.22% Collagenase (C0130, Sigma-Aldrich, Oakville, ON, Canada), 30 μg/ ml DNase (D4263, Sigma-Aldrich, Oakville, ON, Canada), 1% Penicillin-Streptomycin (P/S, Thermofisher Scientific, Waltham, MA, USA), 2.5 mg/ml Dispase II (D4693, Sigma-Aldrich, Oakville, ON, Canada) and 10% Fetal Bovine Serum, FBS (Wisent Inc., Saint-Bruno, QC, Canada), washed in Phosphate Buffered Saline, PBS, pH7.4, and plated on the collagen-coated plates in collecting duct growth medium made up of DMEM adjusted to an osmolarity of 600 mOSM, glucose, NaCl and urea, supplemented with non-essential amino-acids (NEAA, 1%), ITS (1%), mouse Epidermal Growth Factor, EGF (10ng/ml, TonboBiosciences, San Diego, CA), hydrocortisone (50picoM/mL, Sigma-Aldrich), HEPES (1mM), P/S (1%) and 2% FBS. After 5 days in culture, cells were harvested and cell pellets were stored at -80˚C until use.

## Immunoblotting

E17.5 embryonic kidneys of various genotypes were micro-dissected, snap-frozen and stored at -80˚C until use. The embryonic kidney tissues and primary collecting duct cells were lyzed in the whole cell lysate buffer: HEPES (pH 7.5, 50mM), NaCl (150mM), Glycerol (10%), Triton X/H$_2$O (0.5%), MgCL$_2$ (1.5 mM), ethylene glycol-bis(β-aminoethyl ether)-N,N,N′,N′-tetraacetic acid, EGTA, (1mM), Na Fluoride (25mM), sodium orthovanadate (2mM), sodium pyrophosphate (Ppi, 10mM) and protease inhibitor cocktail (PIC, 1X). All chemicals were from Sigma-Aldrich or Thermofisher. 50μg of the lysate per lane were resolved on 8% sodium dodecyl sulfate–polyacrylamide gel, and the proteins were transferred on a nitrocellulose membrane (BioTrace, Menlo Park, CA, USA). The membranes were blocked for 1.5 hours in 5% milk in Tris-buffered saline/Tween 20 buffer, pH 7.4, at room temperature and immuno-probed with goat anti-Vangl2 primary antibody (1:1000 Santa Cruz Biotechnology, Dallas, TX, USA) in the blocking buffer and then with the secondary donkey horseradish peroxidase-conjugated anti-goat antibody (1:10 000, Jackson Immunoresearch, West Grove, PA, USA). The proteins were visualized using enhanced chemiluminescence developing system (West Pico ECL kit, Thermo Scientific, Canada).

## Immunofluorescence staining

For paraffin embedding, all embryonic and postnatal tissues were fixed overnight at 4˚C in 4% paraformaldehyde (PFA)/PBS, pH 7.4, dehydrated with an increasing concentrations of ethanol/PBS and embedded in paraffin blocks (McGill University Health Center Research Institute Histology service). 4 μm sections were made within the same plane in the tissue from the kidneys of all genotypes. For cryo-immunofluorescence, mouse trunks and postnatal kidneys were incubated in 4% PFA/PBS, pH7.4, for 2–4 hours, washed in cold PBS and then incubated sequentially in 15% and 30% sucrose/PBS solutions on a rotating platform until tissue saturation. All tissues were cryo-preserved in the Optimal Cutting Temperature compound (Tissue Tek, Sakura, East Essex, UK) and stored at -80˚C until use. For all experiments, the tissues were freshly cut at 5 μm. For the "tile" confocal microscopy of postnatal kidneys, 30 μm sections were generated.

The paraffin-embedded embryonic and postnatal sections were deparaffinized with xylene at 55˚C and rehydrated in progressively decreasing concentrations of ethanol. The epitope retrieval was performed by immersing the slides in the pre-heated 10nM Na-Citrate buffer (Vector Laboratories, Burlingame, CA, USA) for 20 minutes in a microwave. Special care was taken to ensure that the boiling buffer was covering the sections at all times. Tissues were blocked at room temperature for 40 minutes in a solution of 3% Bovine Serum Albumin (Sigma-Aldrich, Oakville, ON, Canada), 5% Normal Goat Serum (Vector Laboratories,

Burlingame, CA, USA), 5% Normal Donkey Serum (Vector Laboratories, Burlingame, CA, USA), 0.1% Triton X (Thermofisher Scientific, Waltham, MA, USA), and 0.05% Sodium Dodecyl Sulphate (Sigma-Aldrich, Oaksville, ON, Canada) in PBS, pH 7.4. Mouse anti-E-Cadherin (1:20, Invitrogen, *CA*, USA) and rabbit anti-Calbindin (1:500, Sigma-Aldrich, Oakville, ON, Canada) primary antibodies were used following by an incubation with secondary antimouse IgG Alexa 546 and anti-rabbit IgG Alexa 488 antibodies (1:200 and 1: 500, respectively, Invitrogen, Carlsbad, CA, USA). The sections were stained with 4',6-diamidino-2-phenylindole (DAPI, 1:200, Sigma-Aldrich, Oaksville, ON, Canada) to visualize nuclei. Sections were mounted in ProLong Gold Antifade Reagent (Invitrogen, Carlsbad, CA, USA) mounting medium.

Cryosections were fixed in acetone and blocked in 1% BSA, 0.1% TritonX in PBS, pH 7.4. The slides were then stained with fluorescein-labeled Dolichos Biflorus Agglutinin, DBA (1:150, Vector Laboratories Burlingame, CA, USA) or lotus tetragonolobus agglutinin, LTA (1:200, Vector Laboratories, Burlingame, CA, USA) as well as with rabbit anti-Vangl2 antibody (1:20, described in [32]), anti-phosphorylated Myosin Light Chain, MLC (Sigma-Aldrich, Oaksville, ON, Canada) and anti-MLC (Sigma-Aldrich, Oaksville, ON, Canada) antibodies followed by an appropriate secondary antibody. All sections were stained with DAPI (1:200, Sigma-Aldrich, Oakville, ON, Canada) and mounted in ProLong Gold Antifade Reagent or Fluoromount mounting medium (Southern Biotech, Birmingham, AL, USA). For the tile confocal microscopy, the 30 μm kidney sections were stained with rhodamine-conjugated DBA (1:50) and LTA (1:100) for 1h at room temperature. The Z-stack projections of each section were taken on the Zeiss LSM780 microscope, and the images of the entire kidney section was reconstructed by using 'tile' function of the tile scan Zeiss Zen 2.3 software.

For light microscopy, the sections were stained in the Mayer's hematoxylin solution (1:1, MHS16, Sigma-Aldrich, Oakville, ON, Canada) for 4 minutes at room temperature. The slides were rinsed quickly in 95% ethanol and placed in eosin Y alcoholic solution and glacial acetic acid (1:200, HT110116, Sigma-Aldrich, Oakville, ON, Canada). The slides were mounted using a xylene substitute mounting medium (Thermo Shandon Limited, Runcorn Cheshire, UK). Image acquisition was performed using the Zeiss (AxioObserver Z1 inverted fluorescence microscope) using the Zen blue Disk 2012 program.

## Morphological measurements

Quantitative analysis of transverse tubules stained with LTA and DBA was as follows: a measurement of both length and width of each tubular structure was taken. The "length" was defined as the longest diameter in a tubular cross-section, the "width" as perpendicular to it, and the width-to-length ratio was calculated. The area of each round tubule structure was measured by using the "contour" function of the software. DAPI-stained nuclei were counted for each cross-section. 20–30 tubular structures per kidney were assessed; 3 to 4 animals per genotype were used for each age group analyzed. AC analysis was performed on DBA(+) transverse tubule sections of E17.5 embryos, P1 and P7 kidneys of various genotypes by measuring the lengths of the apical and basal surfaces of each cell in a tubule (delineated by the anti-E-Cadherin antibody). The lengths were defined as spanning from one lateral cellular membrane to the other. The apical-to-basal length ratio was calculated. The measurements were conducted on at least 60 transverse tubule sections in each of the 4 animals per genotype.

To assess the localization of apical (luminal side) p-MLC in mutant and control embryonic E17.5 kidneys, we recorded two parameters in the DBA(+) anti-phospho-MLC antibody-stained tubules: the immunofluorescence (IF) pixel intensity at the luminal zone and the pixel intensity in the entire cross-section using Zeiss software. The IF intensity of pMLC was calculated

following the formula $\frac{luminal\ IF\ intensity/\ luminal\ area}{total\ IF\ intensity-luminal\ IF\ intensity\ area}$. A minimum 20 structures per embryo were assessed, 3 embryos per genotype were used.

## Analysis of collecting duct functions

Mice of different ages were stimulated to urinate by bladder massage. The spot urine samples were collected and frozen at -20˚C until use. The measurements of urine $Na^{2+}$ and $K^+$ were serviced to the Comparative Medicine & Animal Resources Centre, McGill University. Urine samples from a minimum of 3 mice of both sexes per age group for each genotype were used. Urine pH was measured by spotting 3µl of each urine sample on the pH paper with 0.3 pH increments. 3–10 samples per age group per each genotype were used for the pH measurements.

## Statistical analysis

To test for significant differences between the groups, the' t-test with a two-tailed distribution and an unequal variance was used. If more than two groups were analyzed, one-way Anova was used. A p-value $< 0.05$ was considered statistically significant. Standard mean errors are shown in each bar graph.

## Results

### Tubule development in *Vangl2*<sup>Lp/Lp</sup> mouse

Homozygous *Vangl2*<sup>Lp/Lp</sup> embryonic kidneys are abnormally shaped and dysplastic [17, 18]. This complex phenotype might implicate several mechanisms, including reduced branching of ureteric bud, abnormal induction of nephron progenitors or defective maturation of tubular segments [4, 33]. We stained embryonic (E) day 17.5 kidneys of wildtype and *Vangl2*<sup>Lp/Lp</sup> mice with anti-calbindin antibody (to visualize the ureteric bud), anti-N-CAM antibody (to detect early nephrogenic structures) or Lotus tetragonolobus (LTA) (to identify proximal tubules). The staining showed no evidence of defective early nephrogenesis (S1 Fig); the number of ureteric buds was not significantly different in *Vangl2*<sup>Lp/Lp</sup> vs wildtype kidneys (S1 Fig). However, we noted profound dilatation of both LTA(+) proximal tubules and calbindin(+) collecting ducts in the *Vangl2*<sup>Lp/Lp</sup> kidneys vs wildtype tissues (Fig 1A).

During embryonic kidney development, tubulogenesis depends on the PCP pathway-controlled CE process; disruption of CE leads to tubular shortening and dilatation [13]. Therefore, we analyzed CE in the E17.5 *Vangl2*<sup>Lp/Lp</sup> and wildtype LTA(+) and calbindin(+) tubules by measuring the cross-sectional area, the diameter along the longest axis (length) and the diameter perpendicular to it (width) as well as the tubular cell number (Fig 1B). Only the tubular cross-sections with the length-to-width ratio within a 30% difference were analysed to avoid structures with overtly elongated shapes. As evidenced by a significantly larger average cross-sectional area and a higher cell number in both the LTA(+) and DBA(+) tubules in the *Vangl2*<sup>Lp/Lp</sup> compared to wildtype kidneys, CE was disrupted in the mutant proximal tubules (Fig 1C) and collecting ducts (Fig 1D); mutant area distribution was wider in range and skewed towards a larger size (Fig 1E). Thus, our results are consistent with a role for *Vangl2* in regulating CE during embryonic tubulogenesis.

### Generation of *Vangl2* mice with ubiquitous and conditional excision of *Vangl2* gene

Homozygous *Lp* mice die *in utero* at E18.5, precluding analysis of postnatal kidneys. We therefore used a *Vangl2* "floxed" allele (*Vangl2*<sup>Fl/Fl</sup>), previously described in Rocque *et al* [20]. The

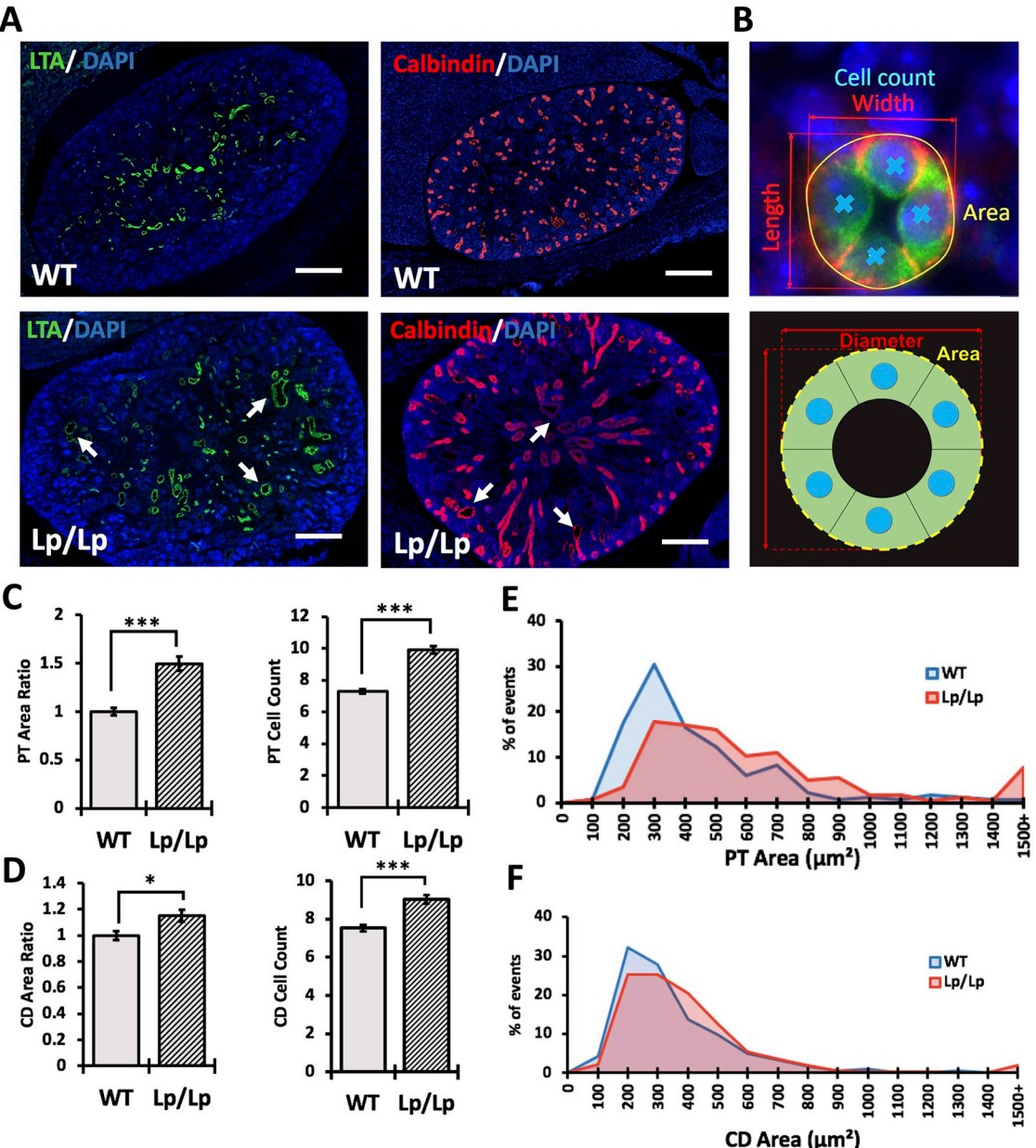

**Fig 1. Kidney phenotype in *Looptail* embryos.** (A) Immunofluorescence staining of E17.5 wildtype and *Looptail* kidney sections with LTA (marker for proximal tubules, green) and 4′,6-diamidino-2-phenylindole (DAPI, marker for nuclei, blue), anti-Calbindin antibody (marker for collecting ducts, red) and DAPI. Scale bars, 100 $\mu$m. (B) Immunostaining (upper image) and schematic representation (lower image) of measurements of area (yellow), diameter (red) and cell count (blue) in tubular cross-sections. (C) Measurements of proximal tubule (PT) area ratio and cell count: WT (n = 181), *Lp/Lp* (n = 280). The average area in wildtype tubules is considered "1" and the ratio of the mutant-to-control tubular area is calculated. (D) Measurements of collecting duct (CD) area ratio and cell count: WT (n = 183), *Lp/Lp* (n = 310). (E) Percentage of events within a given range of cross-sectional area in proximal tubules. (F) Percentage of events within a given range of cross-sectional area in collecting duct tubules. Mean ± standard error of mean is shown in all graphs; p<0.05 (*), p<0.001 (**), p <0.0001 (***).

insertion of a LoxP site into Intron 3 results in an extra 50 bp (Fig 2A and 2B). By crossing the *Vangl2*^Fl/Fl mouse with the *MORE-Cre* mouse expressing Cre recombinase under control of the early embryonic *Meox2* promoter [34], we generated a constitutive *Vangl2* null "Δ" allele (600bp fragment lacking exon 4) vs a 1900bp fragment detectable in *Vangl2*^Fl/Fl animals (Fig

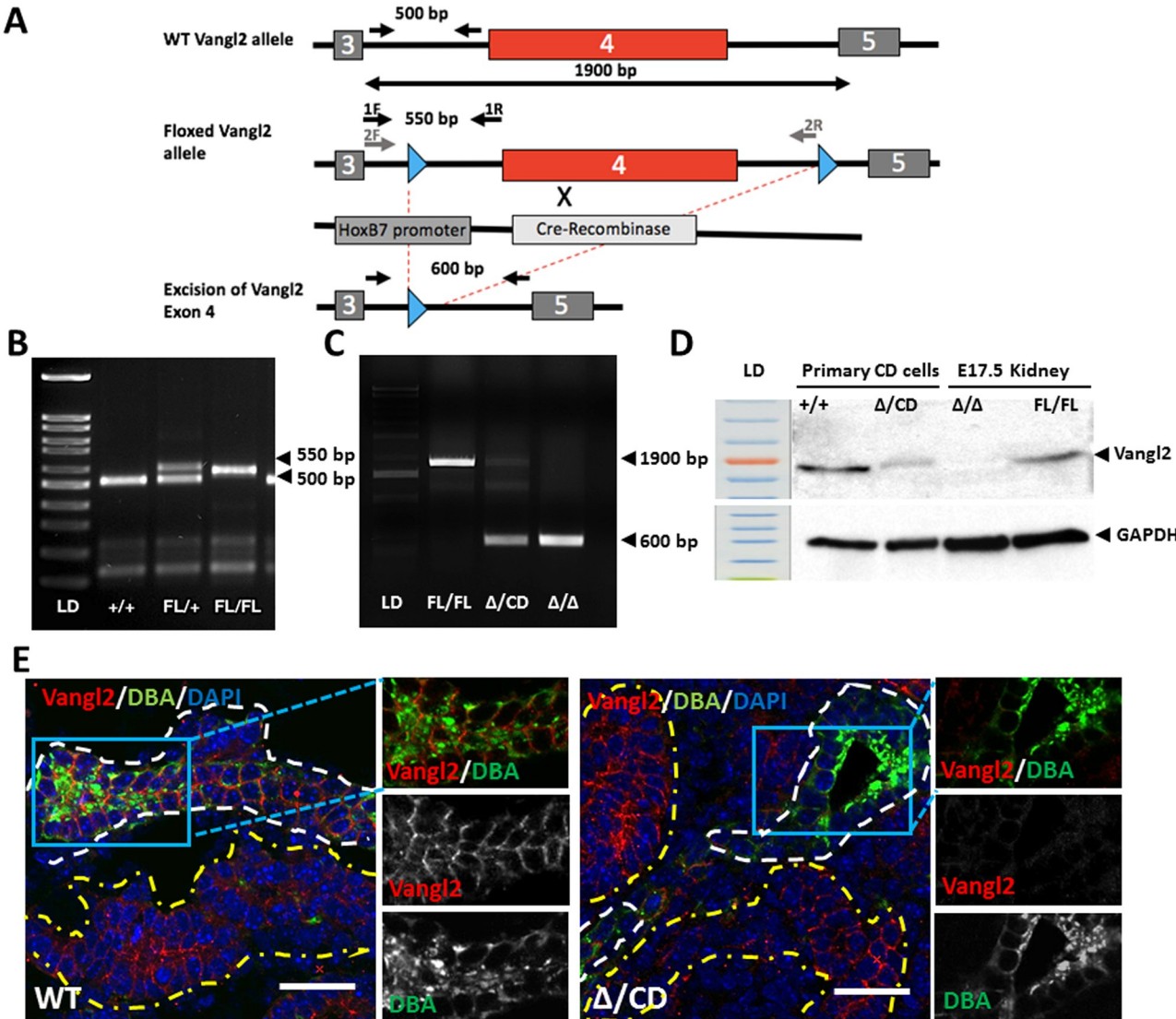

**Fig 2. Generation of *Vangl2* knockout mouse with excision of Exon 4.** (A) Strategy to excise *Vangl2* gene in collecting duct cells using Cre-Recombinase under the *HoxB7* promoter. (B) Images of tail DNA amplicons: 500 bp wildtype and 550 bp Floxed *Vangl2* allele. (C) Excision of *Vangl2* (Δ Exon 4) generates a 600 bp amplification fragment, the Floxed allele is amplified as a 1900 bp fragment; Fl/Fl and Δ/Δ lanes- tail DNA amplification, Δ/CD lane—amplification of DNA from medullary zone of P1 kidneys. (D) Western immunoblot with anti-Vangl2 and anti-GAPDH antibodies: protein lysates from primary collecting duct cells isolated from P5 *Cre+;Vangl2*$^{+/+}$ and *Vangl2*$^{Δ/CD}$ mice as well as protein lysates from E17.5 kidneys of *Cre-;Vangl2*$^{Fl/Fl}$ and *Vangl2*$^{Δ/Δ}$ were used. (E) Immunofluorescent staining with anti-Vangl2 antibody (red), DBA (collecting duct marker, green) and DAPI (blue). DBA(+) tubules are contoured in white, DBA(-) tubules—in yellow. Scale bars, 50 μm.

2C). To study the effect of Vangl2 loss on kidney tubules postnatally, we used *Hoxb7-Cre* mice: Cre-recombinase is driven by the *Hoxb7* promoter expressed exclusively in the collecting duct cells from the onset of metanephric kidney development [29, 35]. Crossing the *Hoxb7-Cre* mouse with the *Vangl2*$^{Δ/Fl}$, we generated *Hoxb7-Cre;Vangl2*$^{Δ/CD}$ (where the "CD" allele is the tissue-specific excision of the *Vangl2* Exon 4 in collecting ducts) (Fig 2A). Reduction of the wildtype allele expression in the DNA isolated from the medullary portion of postnatal (P) day 1 *Vangl2*$^{Δ/CD}$ kidneys (enriched for the collecting duct bundles) confirmed successful excision of Exon 4 in this tissue (Fig 2C, Δ/CD lane). Immunoblotting with anti-Vangl2 antibody [32] showed reduced expression of Vangl2 protein in primary collecting duct cells isolated from P5

renal medulla of $Vangl2^{\Delta/CD}$ mice vs $Vangl2^{Fl/Fl}$ mice (Fig 2D, Δ/CD vs +/+ lane). No Vangl2 protein could be detected in the lysate from $Vangl2^{\Delta/\Delta}$ kidneys (Fig 2D). Vangl2 protein was easily detected by immunofluorescence with anti-Vangl2 antibody in various E17.5 renal tubules (Fig 2E, left panels). However, Vangl2 was undetectable in DBA(+) collecting ducts of $Vangl2^{\Delta/CD}$ kidneys, while adjacent DBA-negative segments continued to express endogenous Vangl2 (Fig 2E, right panel). This confirmed efficient $Vangl2$ gene excision specifically in the collecting duct cells.

## Kidney analysis in $Vangl2^{\Delta/\Delta}$ and conditional $Vangl2^{\Delta/CD}$ mice

To determine whether the "Δ" allele has a fully penetrant kidney phenotype, we generated $Vangl2^{\Delta/\Delta}$ mice. These mice were phenotypically indistinguishable from homozygous $Vangl2^{Lp/Lp}$ mice and died *in utero* at ~E18.5 (Fig 3A). The extent of abnormalities in the E17.5 $Vangl2^{\Delta/\Delta}$ kidneys appeared similar to the $Vangl2^{Lp/Lp}$ mice (Fig 3B and 3C). CE was significantly disturbed in both LTA(+) proximal tubules and Dolichos biflorus agglutinin collecting ducts (DBA, marker of collecting duct cells) (Fig 3D and 3E). The cross-sectional areas in both LTA(+) and DBA(+) tubules were distributed over wider and larger ranges in $Vangl2^{\Delta/\Delta}$ vs $Vangl2^{Fl/Fl}$ tubular structures (Fig 3F and 3G).

At E17.5, kidneys of conditional *Hoxb7-Cre;Vangl2*$^{\Delta/CD}$ mice had abnormal renal morphology, seen as frequently dilated collecting ducts (Fig 4A). Measurements of cross-sectional area and the cell number in the collecting duct cross-sectional structures confirmed defective CE in mutant vs $Vangl2^{Fl/Fl}$ controls, albeit the degree of abnormalities in the $Vangl2^{\Delta/CD}$ was more moderate than in $Vangl2^{\Delta/\Delta}$ kidneys (Figs 3C, 4A, 5A–5D graphs for E17.5 kidneys). This analysis confirmed the autonomous function of the $Vangl2$ gene in controlling tubular area of embryonic collecting ducts and the utility of this conditional mouse.

We next analyzed morphology and CE features in the DBA(+) tubules in the P1, P7, P30, and P90 $Vangl2^{\Delta/CD}$ mice (Figs 4 and 5, S2 and S3 Figs) as well as in the 6 and 9 month-old mutants (S4 Fig). Two controls *Cre(-);Vangl2*$^{Fl/Fl}$ and *Cre(+);Vangl2*$^{+/+}$ were analyzed in parallel for all time points, in order to eliminate any possible phenotypic contribution of the Cre-allele (mixed background). Surprisingly, we did not detect large differences between the mutant vs control kidney phenotypes. The morphological analysis of conditional mutants at P1 showed rare cortical cysts (Fig 4B, S2 Fig) and an otherwise normal renal architecture. At P7, P30 or P90, we did not see morphological defects, albeit some very small differences in the tubule size were detected, mostly between the *Cre(+);Vangl2*$^{\Delta/CD}$ mutant mice and *Cre(-); Vangl2*$^{Fl/Fl}$ controls (Fig 5A and 5B, S2 and S3 Figs), likely reflecting some contribution of the mixed background from the *Cre(+)* allele. The morphological analysis of 6 and 9 month-old mutants did not reveal significant abnormalities in either tubular or glomerular morphology compared to control animals of matching ages (S4 Fig). These results indicate that loss of the $Vangl2$ gene in embryonic kidneys is sufficient to cause tubular dilatation, which appears to be "rescued" shortly after birth. Indeed, analysis of the upper quartile cross-sectional tubular area and cell count at E17.5, P1 and P7 in $Vangl2^{\Delta/CD}$ vs control kidneys revealed a highly significant difference during the embryonic phase, which was reduced yet still significant at P1 and completely disappeared by P7 (Fig 5C and 5D).

Fine-tuning of potassium balance occurs in renal collecting ducts; this is crucial for proper whole-body potassium homeostasis [36]. We measured urinary potassium/sodium ratio and urinary pH as a measure of collecting duct apical Na/K or Na/H+ exchange in P30, P90, 6 month and 9 month-old $Vangl2^{\Delta/CD}$ mice and in both control mouse lines (combined data). We detected no significant difference in collecting duct urinary Na/K ratio between any of the

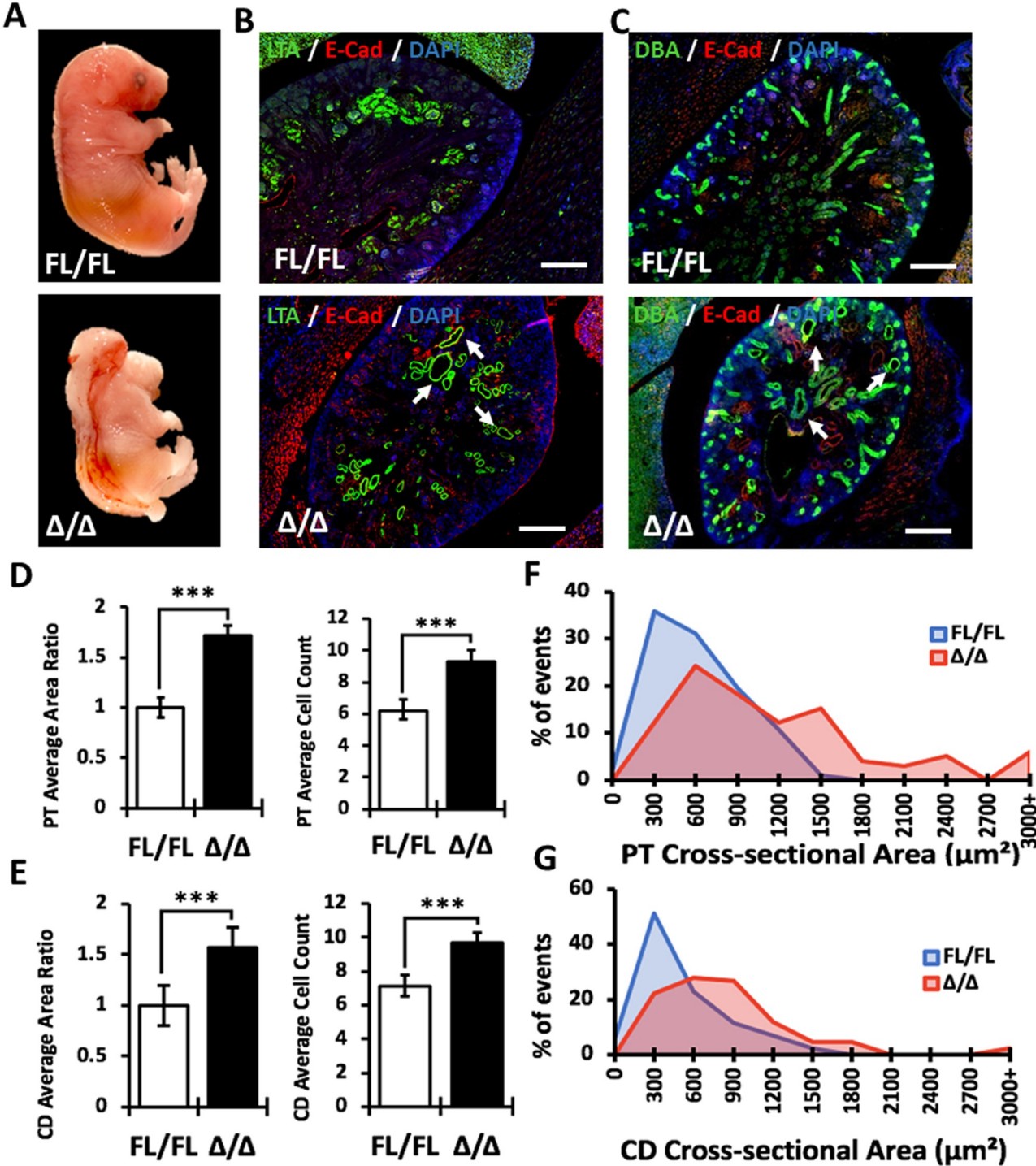

**Fig 3. Analysis of kidney morphology in null E17.5 *Vangl2*^Δ/Δ embryos.** (A) E17.5 *Vangl2*^Fl/Fl (upper image) and null *Vangl2*^Δ/Δ (lower image) embryos. (B) Immunofluorescence staining of E17.5 *Vangl2*^Fl/Fl and *Vangl2*^Δ/Δ kidney sections with LTA (green), E-cadherin (marker for lateral cell connections, red) and DAPI (blue); cystic and dilated proximal tubules are indicated by arrows. Scale bars, 100 $\mu$m. (C) Immunofluorescence staining of E17.5 *Vangl2*^Fl/Fl and *Vangl2*^Δ/Δ kidney sections with DBA (green), E-cadherin (red) and DAPI (blue); cystic and dilated collecting ducts indicated by arrows. Scale bars, 100 $\mu$m. (D) Measurements of proximal tubule (PT) area ratio and cell count: *Vangl2*^Fl/Fl (n = 80), *Vangl2*^Δ/Δ (n = 77). (D) Measurements of collecting duct (CD) area ratio and cell count: *Vangl2*^Fl/Fl (n = 88), *Vangl2*^Δ/Δ (n = 86). (E) Percentage of events within a given range of tubular cross-sectional area of proximal tubules. (F). Percentage of events within a given range of tubular cross-sectional area of collecting duct tubules. Mean ± standard error of mean is shown in all graphs. $p < 0.05$ (*), $p < 0.001$ (**), $p < 0.0001$ (***).

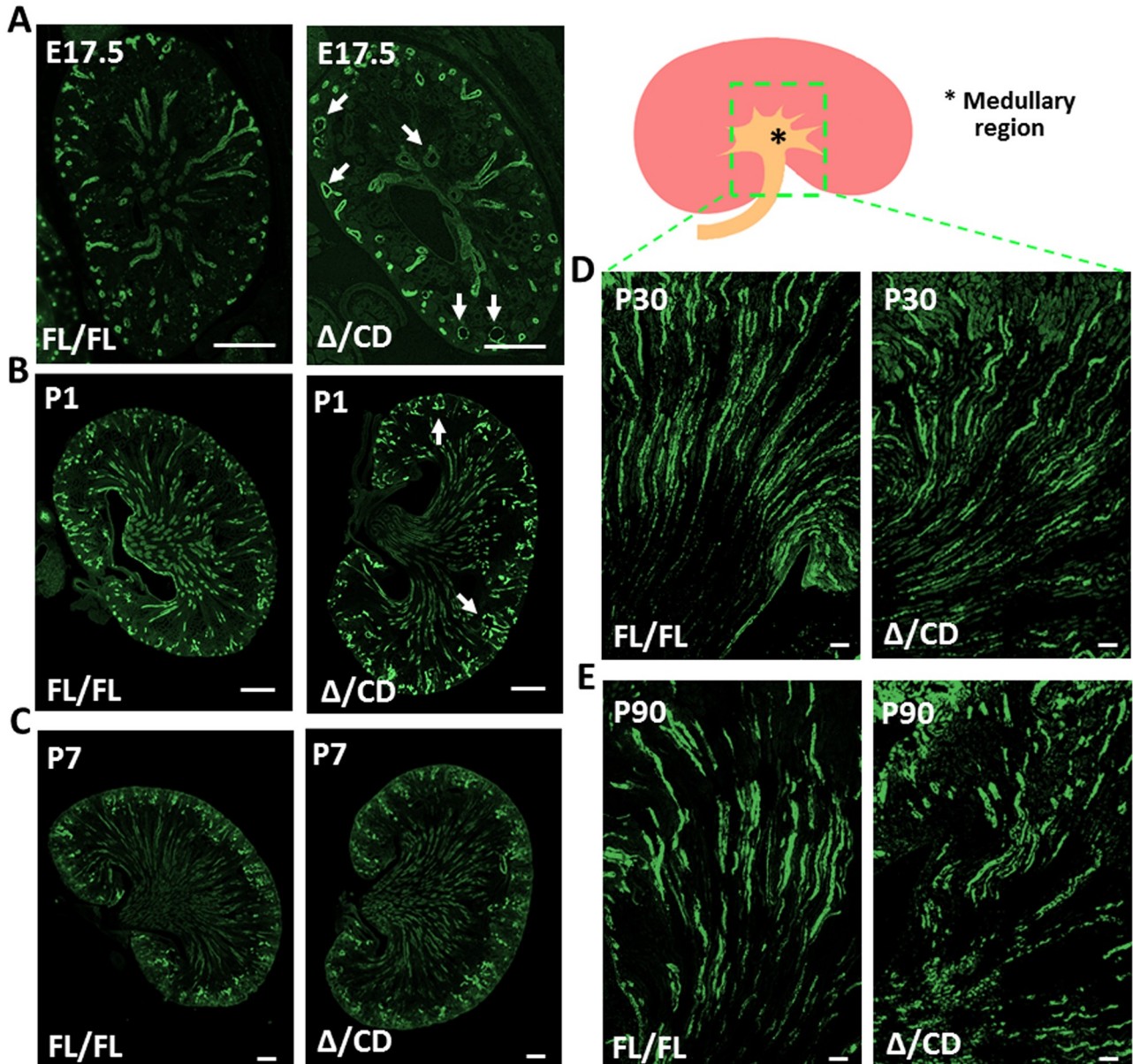

**Fig 4. Morphology of collecting duct tubules in conditional Hoxb7-Cre;*Vangl2*$^{\Delta/CD}$ mice.** (A) E17.5 *Vangl2*$^{Fl/Fl}$ (control) and *Vangl2*$^{\Delta/CD}$ (mutant) kidneys stained with DBA (collecting duct marker, green); dilated tubules are indicated by arrows. Scale bars, 100 $\mu$m. (B) Postnatal day 1 (P1) *Vangl2*$^{Fl/Fl}$ and *Vangl2*$^{\Delta/CD}$ kidneys stained with DBA (green); dilated tubules are indicated by arrows. Scale bars, 100 $\mu$m. (C) P7 *Vangl2*$^{Fl/Fl}$ and *Vangl2*$^{\Delta/CD}$ kidneys stained with DBA (green); Scale bars, 100 $\mu$m. (D) P30 kidney medullary region stained with DBA (green), Scale bars, 100 $\mu$m. (E) P90 kidney medullary region stained with DBA (green), Scale bars, 200 $\mu$m.

groups (Fig 5E and 5F). Although urine pH was marginally higher in the conditional mutants at 9 months, it was similar among all groups at all other time points.

We also examined conditional *Vangl2* mutant mice with excision of the *Vangl2* gene driven by the Pax2-Cre promoter, *Pax2-Cre;Vangl2*$^{\Delta TM/LoxP}$ mice [31]; the *Pax2-Cre*-driven gene excision occurs along the entire nephron [37]. We found no morphological abnormalities in P1 or P15 in the *Pax2-Cre; Vangl2*$^{\Delta TM/LoxP}$ kidneys (S5 Fig).

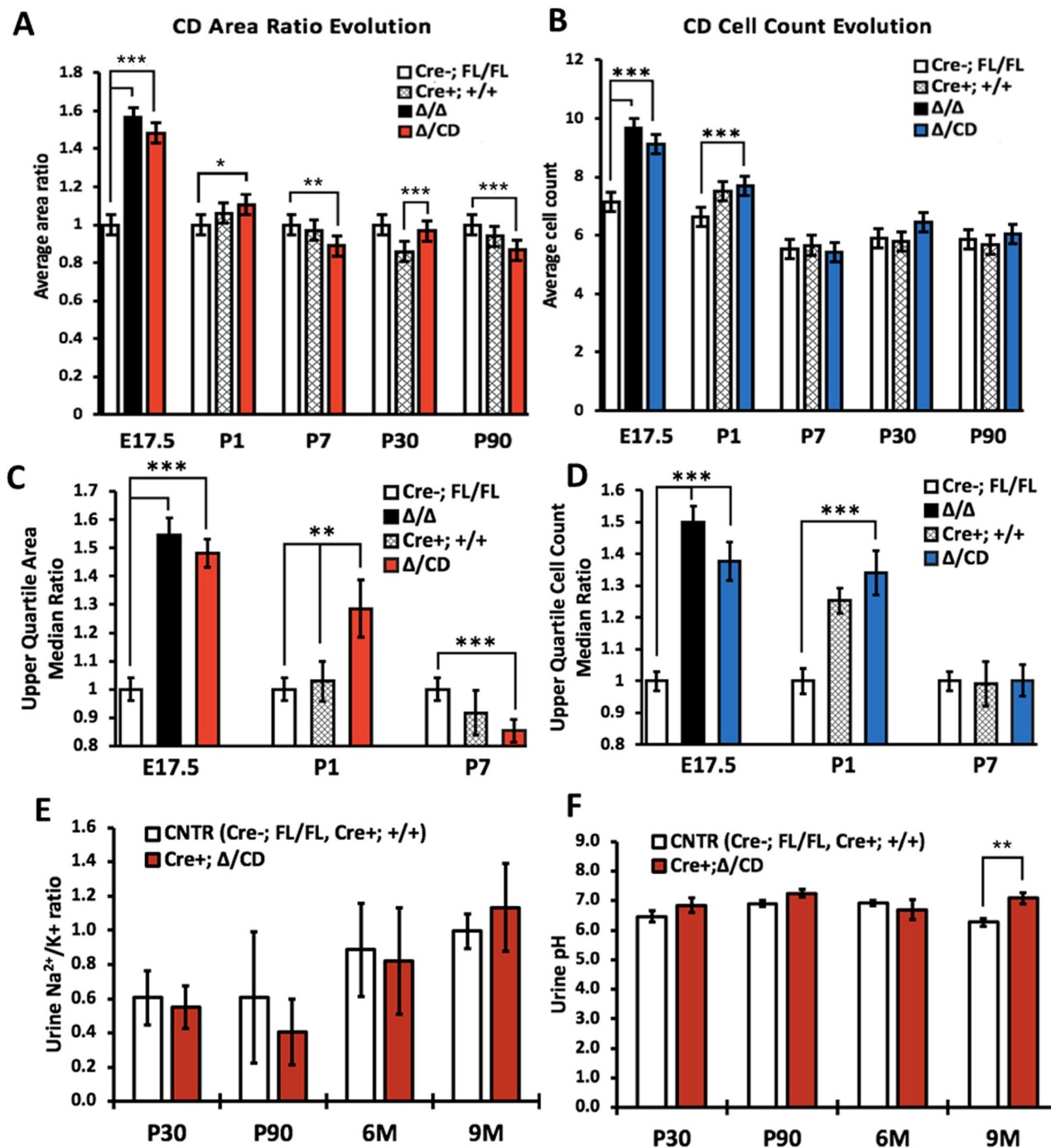

**Fig 5. Quantitative analysis of the collecting duct morphological features and function in *Vangl2* mutant mice.** (A) Longitudinal measurement of collecting duct cross-sectional area ratio in E17.5—P90 mice: ubiquitous mutants *Vangl2*$^{\text{Lp/Lp}}$ (n = 310), *Vangl2*$^{\Delta/\Delta}$ (n = 108), conditional mutant *Vangl2*$^{\Delta/CD}$ (E17.5 n = 93, P1 n = 362, P7 n = 163, P30 n = 221, P90 n = 117), *Cre(-);Vangl2*$^{\text{Fl/Fl}}$ (E17.5 n = 93, P1 n = 324, P7 n = 194, P30 n = 245, P90 n = 128) and *Cre(+);Vangl2+/+* (P1 n = 337, P7 n = 109, P30 n = 212, P90 n = 141) (two latter are controls). (B) Longitudinal measurement of collecting duct cross-sectional cell counts in the same structures as in (A). (C) The upper quartile area ratio in the collecting duct tubules in ubiquitous mutant *Vangl2*$^{\Delta/\Delta}$ (E17.5 n = 20), conditional *Vangl2*$^{\Delta/CD}$ (E17.5 n = 27, P1 n = 78, P7 n = 41), mutants and *Cre(-);Vangl2*$^{\text{Fl/Fl}}$ (E17.5 n = 22, P1 n = 85, P7 n = 46) and *Cre(+);Vangl2+/+* (P1 n = 83, P7 n = 25) at E17.5, P1 and P7. (D) The upper quartile cell counts in the collecting duct tubule structures used in (C). (E) Urine K$^+$/Na$^{2+}$ ratio in P30-9 month-old conditional *Vangl2*$^{\Delta/CD}$ and control *Cre(+) Vangl2*$^{+/+}$ and *Cre(-);Vangl2*$^{\text{Fl/Fl}}$ mice: minimum 3 specimens per time point per genotype were used. (F) Urine pH in P30–9 month-old conditional *Vangl2*$^{\Delta/CD}$ and control *Cre(+) Vangl2*$^{+/+}$/ *Cre(-); Vangl2*$^{\text{Fl/Fl}}$ mice: minimum 6 specimens per time point per genotype were used. The data represent mean ± standard error of mean in all graphs. p<0.05 (*), p<0.01 (**), p <0.001 (***).

## Apical constriction in the embryonic renal tubules lacking *Vangl2*

Packaging of cells into the tubular wall requires cell shape remodeling known as apical constriction (AC), a process thought to rely on PCP signaling [26, 27]. We analyzed AC in DBA (+) collecting duct cross-sectional tubules at E17.5, P1 and P7. All tissues were co-stained with anti-E-cadherin antibody, to delineate the lateral borders of each cell; the apical and basal lengths were measured (as shown in Fig 6A), and the apical-basal length ratio was calculated (Fig 6B and 6C). We observed a significantly larger apical-basal ratio in $Vangl2^{Lp/Lp}$ or $Vangl2^{\Delta/\Delta}$ embryos compared to $Vangl2^{Fl/Fl}$, indicating that the cells from the mutants with ubiquitous loss of *Vangl2* gene were more cuboidal (Fig 6B). The AC phenotype in DBA(+) tubular cells of E17.5 conditional $Vangl2^{\Delta/CD}$ mouse was milder than in the null mutants, yet still significantly different compared to controls (Fig 6C). At P1, the apical-to-basal length ratio in $Vangl2^{\Delta/CD}$ DBA(+) cells was significantly different from controls. However, no differences in $Vangl2^{\Delta/CD}$ DBA(+) cells were observed at P7, paralleling our observations of infrequent tubular dilation at P1 and a lack of it at P7.

AC is driven by activated actomyosin at the apical cell surface. Activity of actomyosin is controlled by regulatory myosin light chain, MLC, which is phosphorylated (activated) in response to various stimuli [38](Fig 6E). Using an antibody which detects phospho-MLC (pMLC), we measured pMLC immunofluorescence intensity to determine whether activated MLC is enriched at the apical surface. We observed asymmetric accumulation of pMLC along the luminal surface of the control $Vangl2^{Fl/Fl}$ DBA(+) collecting duct cells, indicative of active AC process (Fig 6D and 6F). On the contrary, we detected a significant decrease of pMLC intensity at the apical cellular side in the DBA(+) tubules of E17.5 $Vangl2^{\Delta/\Delta}$ (Fig 6D and 6F), consistent with a loss of pMLC accumulation at the apical surface in mutant cells.

## Discussion

In this study, we analyze the kidney phenotype of several mutant mouse models with loss of the key PCP gene *Vangl2*. We show that during the embryonic period, *Vangl2* controls renal tubulogenesis by regulating CE and AC. This might suggest that disturbances of PCP signaling could cause congenital kidney malformations falling into the CAKUT spectrum. Unexpectedly, however, loss of *Vangl2* appears to be dispensable for mature renal architecture.

Yates *et al* and our group previously reported kidney dysplasia in embryonic kidneys of the *Lp* mutant [17, 18]. Loss of the *Vangl2* gene leads to a wider range of tubular sizes in embryonic kidneys indicating that *Vangl2* controls the tubular size during renal development. Our data show that loss of *Vangl2* in embryonic kidneys disrupts CE. These results are consistent with Carroll and colleagues who found that tubular diameter in developing kidneys is regulated by CE rather than oriented cell division (OCD), since mitotic spindles become aligned along the renal tubular plane only in the perinatal period [13]. Others have postulated that the PCP pathway governs OCD and, when deregulated, may cause cyst formation [14]. Kunimoto *et al* recently demonstrated off-axis cell divisions in the kidney tubules of neonatal PCP mutants, however, this did not lead to cyst appearance [39].

We were surprised to find normal tubular architecture in the collecting duct tubules in young or mature $Vangl2^{\Delta/CD}$ mice, and did not detect abnormalities of collecting duct function postnatally. In keeping with our observations, Kunimoto *et al* showed that at 16 weeks of age, tubular diameter in conditional double *Vangl1/Vangl2* mutants was only slightly dilated compared to control and no cysts were found [39]. We cannot attribute the lack of phenotype in postnatal mutant kidneys to the insufficient excision of *Vangl2*; on the contrary, we detected robust excision of *Vangl2* gene driven by *Hoxb7-Cre* and observed tubular dilatation in conditional mice at E17.5 and P1, confirming loss of *Vangl2* gene in collecting duct tubules in our

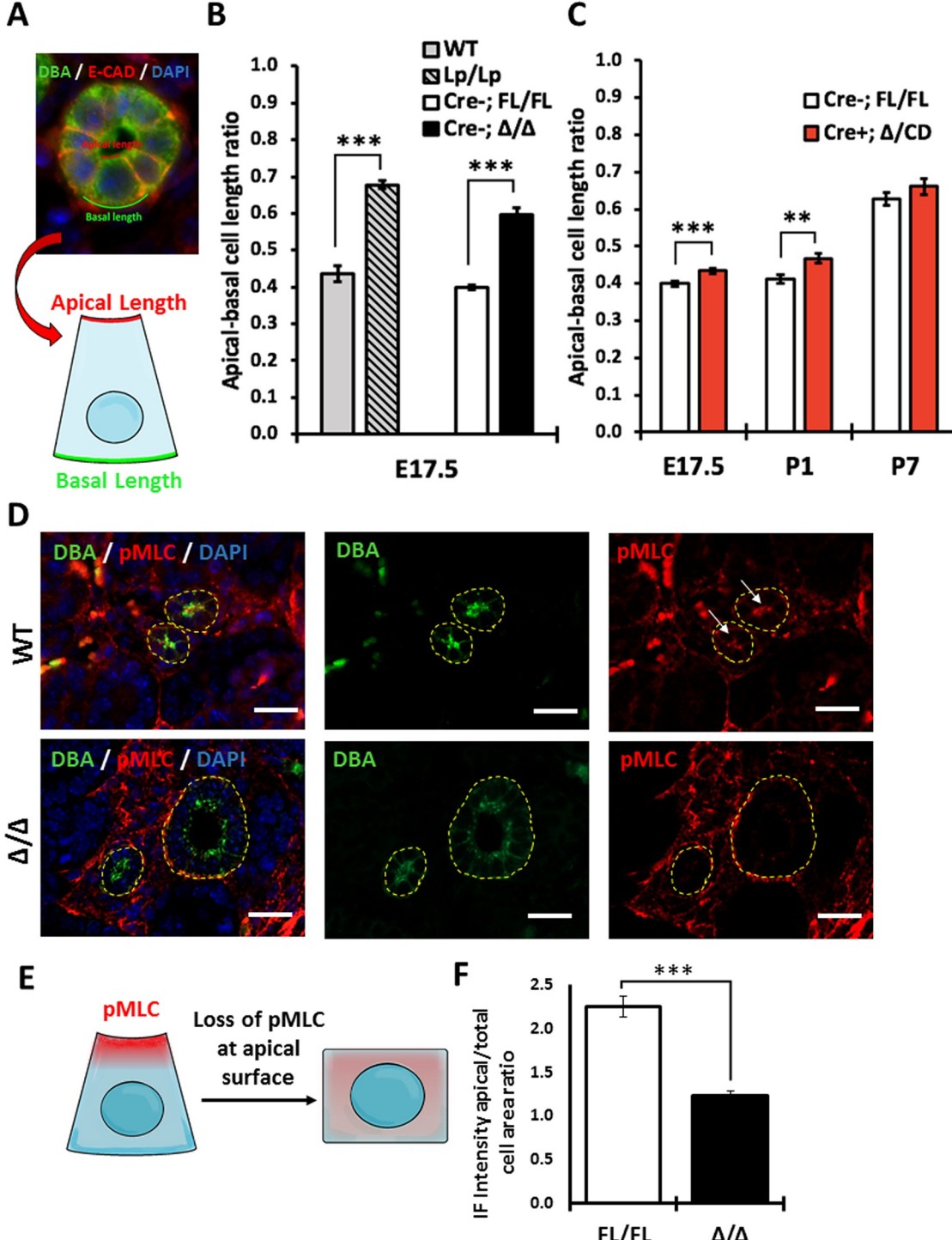

**Fig 6. Analysis of apical constriction in collecting duct of *Vangl2* mutant mice.** (A) Immunofluorescence image (upper panel) and schematic representation (lower drawing) of the AC measurements; DBA (green), E-cadherin (red), DAPI (blue). (B) Measurements of the apical-to-basal-surface ratio in the E17.5 DBA(+) cross-sectional tubules of *Vangl2*$^{Lp/Lp}$ (n = 456), *Vangl2*$^{Δ/Δ}$ (n = 86), wildtype (n = 58) and *Cre(-);Vangl2*$^{Fl/Fl}$ (n = 542) mice. (C) Measurements of the apical-to-basal-surface ratio in E17.5, P1 and P7 DBA(+) cross-sectional tubules of *Vangl2*$^{Δ/CD}$ (E17.5 n = 525; P1 n = 107, P7 n = 82), and *Cre(-);Vangl2*$^{Fl/Fl}$ (E17.5 n = 542; P1 n = 96, P7 n = 143) mice. (D) Immunofluorescence staining of E17.5 wildtype and Vangl2$^{Δ/Δ}$ kidney sections with DBA (green, denoted in yellow line), anti-phospho-Myosin Light Chain antibody (pMLC, red) and DAPI (blue). Arrows point at the luminal pMLC enrichment in control tubules. Scale bars, 100 $\mu$m. (E) Schematic representation of pMLC localization in control and Vangl2

mutant tubule cells. (F) Quantitative analysis of pMLC immunofluorescence intensity in the apical area over the total cell intensity area in $Vangl2^{\Delta/\Delta}$ (n = 100) vs $Vangl2^{Fl/Fl}$ (n = 98). The data represent mean ± standard error of mean in all graphs. p<0.05 (*), p<0.01 (**), p <0.0001 (***).

model. The statistically significant differences between age-matched $Cre-;Vangl2^{Fl/Fl}$ controls and conditional mutants postnatally are most likely driven by background variation, where pure Bl6 mice vary from mice with a mixed background due to the Cre-allele, despite morphologically being similar. Thus, our study and that of Axelrod's group [39] demonstrate that loss of PCP signaling does not lead to major changes in mature tubular architecture and does not cause renal cysts in adult animals.

While embryonic mutant collecting ducts are profoundly dilated, postnatal $Vangl2$ mutant collecting ducts look indistinguishable from the controls. The embryonic tubular dilatation rapidly disappears in the postnatal period. At P1, there is some residual tubular dilatation, but it is fully resolved by P7. This coincides with the period of tubular elongation and rapid renal tubular cell proliferation until ~ P10-12, when there is an abrupt shift in gene expression toward maturational transcriptional programs [40]. Our observations suggest that there is a shift in the mechanisms controlling tubular size from a PCP-dependent embryonic control of tubular diameter to an alternative Vangl2-independent mechanism in the early postnatal period, which is able to resolve earlier abnormalities (Fig 7).

Intriguingly, Deans and colleagues described a similar mechanistic "switch" from embryonic PCP signaling to other unknown regulatory pathways in the developing inner ear in mice [31]. At E18.5, $Vangl2^{Lp/Lp}$ and conditional $Pax2-Cre;Vangl2^{\Delta TM/LoxP}$ mice exhibited profound

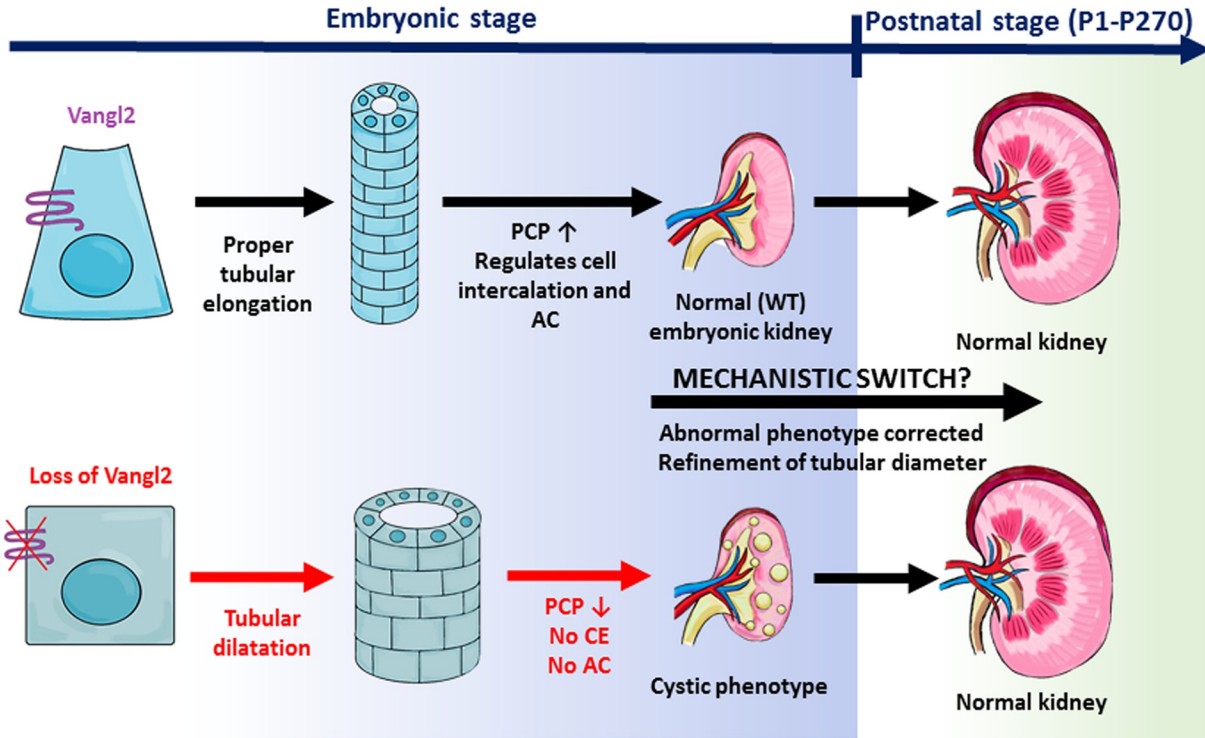

**Fig 7. Role in Vangl2 in embryonic versus postnatal kidney tubules.** During embryonic kidney development, normal PCP signaling in wildtype kidney regulates tubular size via controlling CE and AC. This keeps tubular size within a narrow range. In the embryonic kidneys of *Vangl2* and other PCP mutants, the control of CE and AC is deregulated, resulting in the tubular dilatation and formation of microcysts. After birth, the tubular dilatation phenotype is refined by an unknown PCP-independent mechanism.

defects in the PCP-controlled alignment of stereocilia on sensory hair cells of the organ of Corti. Remarkably, realignment of stereocilia to an orientation similar to littermate controls was observed at 4 days postnatally in conditional mice, demonstrating a rescue of the PCP derangement. It appears that the mechanistic switch from the PCP-controlled processes during development to the PCP-independent mechanisms controlling tissue homeostasis after birth may be a normal widespread occurrence.

It has been inferred that cysts arise in polycystic kidney disease because of faulty PCP signaling [14, 15]. Ohata *et al* identified genetic interaction between *Vangl2* and *Pkd1* in the alignment of multiciliated patches on ventricular brain cells [41]; both Pkd1 and Pkd2 influence asymmetric distribution of the Vangl2 protein in these cells [41]. However, we found no renal cysts in mice with collecting duct excision of *Vangl2*. Furthermore, there is no evidence that PCP-dependent OCD is disturbed prior to cyst appearance in mouse mutants with excision of *Pkd1* or *Pkd2* in renal tubules (genes mutated in autosomal dominant polycystic kidney disease) [42]. Thus, while the biallelic loss of *Pkd1* or *Pkd2* clearly induces cystogenesis in renal tubules, this does not appear to involve defective PCP signaling.

We acknowledge several limitations in the current study. We studied mice with excision of Vangl2, while a paralogous gene, Vangl1, remained intact. Vangl1 and Vangl2 genes have overlapping expression patterns in some but not all tissues [32], are expressed in the renal tubules [32, 39], and interact genetically during formation of neural tube [43] and in controlling planar polarization of primary cilium [44]. Our work revealed that loss of Vangl2 alone is sufficient to bring about abnormal kidney embryonic phenotype, however, as reported by Kunimoto *et al*, simultaneous excision of both Vangl1 and Vangl2 genes in renal tubules may cause a very mild non-cystogenic tubular dilation in adult mice [39], the phenotype which was not observed in our $Vangl2^{\Delta/CD}$ mice. Thus, it is possible that lack of tubular dilation in $Vangl2^{\Delta/CD}$ adult mice was due to the presence of intact Vangl1. Finally, the severity of the tubular phenotype in E17.5 kidneys from the $Vangl2^{CD/\Delta}$ mouse (tissue specific excision of *Vangl2* gene) is milder than the phenotype in $Vangl2^{\Delta/\Delta}$ or $Vangl2^{Lp/Lp}$ mice (ubiquitous loss of *Vangl2* gene function). This may likely be due to a somewhat mosaic Cre-recombinase-driven excision of the *Vangl2*.

In summary, we have generated and analyzed mouse models with embryonic and postnatal loss of the key PCP gene *Vangl2*. We find that *Vangl2* contributes significantly to renal tubulogenesis during embryonic development via controlling CE and AC. Yet its loss is dispensable and non-cystogenic for early postnatal kidney development and renal tubular maintenance and functions in adult animals.

## Supporting information

**S1 Fig. Loss of Vangl2 does not disturb early nephrogenesis.** (A). Kidney development can be visualized using kidney sections. Examination of early nephrogenic structures such as renal vesicles (RVs), comma-shaped bodies (CSB) and S-shaped bodies (SSB) allows to define whether there are any defects in early nephrogenesis. These structures give rise to the glomerulus, proximal and distal tubules. RVs, CSBs and SSBs were detected with anti-NCAM antibody (green, white arrows). The collecting duct is derived from ureteric bud (UB) through repetitive ureteric bud branching. Each UB tip gives rise to a single nephron. The UB tips were visualized with anti-calbindin antibody (red, yellow arrows) in the E17.5 $Vangl2^{Lp/Lp}$ and wildtype E17.5 embryos. (B). The ratio of RV to SCB to SSB as well as the UB number were counted in the maximal kidney cross-sections; minimum 2 cross-sections per embryo per genotype were analyzed, 4 embryos per each genotype were assessed. ~ 300 nephrogenic structures and ~300 UB tips per genotype were analyzed.
(PDF)

**S2 Fig. Morphological analysis of postnatal P1 and P7 kidneys of conditional Hoxb7-Cre;** *Vangl2*$^{\Delta/CD}$ **mice compared to controls.** (A) P1 *Vangl2*$^{Fl/Fl}$ (control) and Cre(+);Δ/CD (*Vangl2*$^{\Delta/CD}$ mutant) renal cortices and pelvic zones stained with hematoxylin and eosin (H&E); dilated tubules and cystic structures are indicated by black arrows, glomeruli are indicated by yellow arrows. Scale bars, 200 $\mu$m. (B) P7 *Vangl2*$^{Fl/Fl}$ and Cre(+);Δ/CD renal cortices and pelvic zones stained with H&E; glomeruli are indicated by yellow arrows. Scale bars, 200 $\mu$m. (C) Percentage of events within a given range of cross-sectional area in collecting duct tubules in P1 *Vangl2*$^{\Delta/CD}$ (n = 362), *Cre(-);Vangl2*$^{Fl/Fl}$ (n = 324) and *Cre(+);Vangl2+/+* (n = 337). (D) Percentage of events within a given range of cross-sectional area in collecting duct tubules in P7 *Vangl2*$^{\Delta/CD}$ (n = 163), *Cre(-);Vangl2*$^{Fl/Fl}$ (n = 194) and *Cre(+);Vangl2+/+* (n = 109). At least 3 animals per genotype were analyzed.
(PDF)

**S3 Fig. Morphological analysis of postnatal P30 and P90 kidneys of conditional Hoxb7-Cre;***Vangl2*$^{\Delta/CD}$ **mice compared to controls.** (A) P30 *Vangl2*$^{Fl/Fl}$ (control) and Cre(+);Δ/CD (*Vangl2*$^{\Delta/CD}$ mutant) renal cortices and pelvic zones stained with H&E; glomeruli are indicated by yellow arrows. Scale bars, 200 $\mu$m. (B) P90 *Vangl2*$^{Fl/Fl}$ and Cre(+);Δ/CD renal cortices and pelvic zones stained with H&E; glomeruli are indicated by yellow arrows. Scale bars, 200 $\mu$m. (C) Percentage of events within a given range of cross-sectional area in collecting duct tubules in P30 *Vangl2*$^{\Delta/CD}$ (n = 221), *Cre(-);Vangl2*$^{Fl/Fl}$ (n = 245) and *Cre(+);Vangl2+/+* (n = 212). (D) Percentage of events within a given range of cross-sectional area in collecting duct tubules in P90 *Vangl2*$^{\Delta/CD}$ (n = 117), *Cre(-);Vangl2*$^{Fl/Fl}$ (n = 128) and *Cre(+);Vangl2+/+* (n = 141). At least 3 animals per genotype were analyzed.
(PDF)

**S4 Fig. Morphological analysis of 6 month-old and 9 month-old kidneys of conditional Hoxb7-Cre;***Vangl2*$^{\Delta/CD}$ **mice, compared to controls.** (A) 6 month-old *Vangl2*$^{Fl/Fl}$ (control) and Cre(+);Δ/CD (*Vangl2*$^{\Delta/CD}$ mutant) renal cortices and pelvic zones stained with H&E; glomeruli indicated by yellow arrows. Scale bars, 200 $\mu$m. (B) P270 *Vangl2*$^{Fl/Fl}$ and *Vangl2*$^{\Delta/CD}$ renal cortices and pelvic zones stained with H&E; glomeruli are indicated by yellow arrows. Scale bars, 200 $\mu$m. At least 3 animals per genotype were analyzed.
(PDF)

**S5 Fig. Morphological analysis of postnatal kidneys of conditional Pax2-Cre;Vangl2 mouse with Vangl2 excision in renal tubules.** This mouse model was originally described in Copley *et al*, 2013. "ΔTMs" allele has a ubiquitous excision of *Vangl2* exon 4 which encodes four transmembrane domains, TMs. The "LoxP" allele is the conditional excision of *Vangl2* exons 2 and 3 encoding ATG translation initiation site. When bred to the *Pax2-Cre* deleter mouse, Cre-recombinase expressed under Pax2 promoter drives excision of *Vangl2* exons 2 and 3 along the entire nephron. Medullary zone is demarcated by yellow line. The morphological analysis by H&E staining and light microscopy revealed no changes in the kidney architecture; no tubular dilatation or cysts were observed. 4 embryos per genotype were examined.
(PDF)

**S1 Checklist. The ARRIVE guidelines checklist.**
(PDF)

**S1 Raw images.**
(PDF)

## Acknowledgments

We would like to thank Carlton Bates for providing HoxB7-Cre mouse. This study was supported by the grants from the Kidney Foundation of Canada KFOC1518 & KFOC1719 and the Canadian Institute of Health Research CIHR230929 to E.T.

## Author Contributions

**Conceptualization:** Philippe Gros, Elena Torban.

**Data curation:** Ida Derish, Jeremy K. H. Lee, Melanie Wong-King-Cheong, Sima Babayeva, Jillian Caplan, Vicki Leung, Chloe Shahinian, Michel Gravel, Elena Torban.

**Formal analysis:** Ida Derish, Jeremy K. H. Lee, Melanie Wong-King-Cheong, Sima Babayeva, Jillian Caplan, Vicki Leung, Chloe Shahinian, Michel Gravel, Elena Torban.

**Funding acquisition:** Philippe Gros, Elena Torban.

**Investigation:** Ida Derish, Sima Babayeva.

**Methodology:** Elena Torban.

**Resources:** Michael R. Deans, Elena Torban.

**Supervision:** Michael R. Deans, Philippe Gros, Elena Torban.

**Validation:** Michel Gravel.

**Writing – original draft:** Ida Derish, Philippe Gros, Elena Torban.

**Writing – review & editing:** Michael R. Deans, Philippe Gros, Elena Torban.

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
