## [Decision Letter · Decision Letter 0]

2 Jan 2020

PONE-D-19-31683

Differential Role of Planar Cell Polarity Gene Vangl2 in Embryonic and Adult Mammalian Kidneys

PLOS ONE

Dear Dr Torban,

Thank you for submitting your manuscript to PLOS ONE. After careful consideration, we feel that it has merit but does not fully meet PLOS ONE’s publication criteria as it currently stands. Therefore, we invite you to submit a revised version of the manuscript that addresses the points raised during the review process.

We would appreciate receiving your revised manuscript by Feb 16 2020 11:59PM. To enhance the reproducibility of your results, we recommend that if applicable you deposit your laboratory protocols in protocols.io, where a protocol can be assigned its own identifier (DOI) such that it can be cited independently in the future. For instructions see: http://journals.plos.org/plosone/s/submission-guidelines#loc-laboratory-protocols

We look forward to receiving your revised manuscript.

Kind regards,

Shree Ram Singh, Ph. D.

Academic Editor

PLOS ONE

Journal Requirements:

2. To comply with PLOS ONE submissions requirements, please provide method(s) of sacrifice in the Methods section of your manuscript.

3. As part of your revision, please complete and submit a copy of the ARRIVE Guidelines checklist, a document that aims to improve experimental reporting and reproducibility of animal studies for purposes of post-publication data analysis and reproducibility: https://www.nc3rs.org.uk/arrive-guidelines. Please include your completed checklist as a Supporting Information file. Note that if your paper is accepted for publication, this checklist will be published as part of your article.

4. Thank you for including the following funding information;"The funders had no role in study design, data collection and analysis, decision to publish, or preparation of the manuscript."

Please provide an amended Funding Statement that declares *all* the funding or sources of support received during this specific study (whether external or internal to your organization) as detailed online in our guide for authors at http://journals.plos.org/plosone/s/submit-now.  

Please state what role the funders took in the study.  If any authors received a salary from any of your funders, please state which authors and which funder. If the funders had no role, please state: "The funders had no role in study design, data collection and analysis, decision to publish, or preparation of the manuscript."

Reviewers' comments:

Reviewer's Responses to Questions

**Comments to the Author**

1. Is the manuscript technically sound, and do the data support the conclusions?

Reviewer #1: Yes

Reviewer #2: Yes

2. Has the statistical analysis been performed appropriately and rigorously? 

Reviewer #1: No

Reviewer #2: Yes

3. Have the authors made all data underlying the findings in their manuscript fully available?

Reviewer #1: Yes

Reviewer #2: Yes

4. Is the manuscript presented in an intelligible fashion and written in standard English?

Reviewer #1: Yes

Reviewer #2: Yes

5. Review Comments to the Author

Reviewer #1: The study by Ida Derish and co-workers tried to elucidate the role of the key PCP gene, Vangl2, in embryonic and postnatal renal tubules and ascertain whether its loss contributes to cyst formation and defective tubular function in mature animals. The study was well conducted and well written and the results obtained are interesting. There are few points and comments which needs to be addressed prior to decision by the Editor

Major points:

1. HoxB7-Cre mice used were on a mixed B16/CD-1 background. What is their cell/tissue expression pattern?

2. Related HoxB7-Cre mice seems to show low levels of expression in the dorsal root ganglia and the spinal cord? How did the authors address the issue?

3. Genotyping by visual inspection is not an ideal method, Due to variation in phenotypes. Proper PCR based Genotyping should be used.

4. Immunofluorescence staining: Why was the thickness 30micrometer chosen for tile confocal? Does the author used 3D scan?

5. Statistical analysis: Does all the data pass Normality and Equal Variance test?

6. Figure 5A: P30, & Figure 6C: E17.5, Statistical significance need to be analyzed again.

7. It’s good to add some limitations for the study in the discussion so that the reader is able to see where the gaps are for this article and which the future work can focus on.

Minor points:

1. The manuscript should have line numbers, without which it is difficult to point out the errors.

2. The Words ”Convergent extension and apical constriction” can be abbreviated at its first mention in Abstract.

3. Convergent extension (CE) and apical constriction (AC) were abbreviated in Introduction itself. Hence from then on the abbreviation alone is sufficient.

4. The words “Sibling mating” is better than “Brother-Sister mating”

5. Animal breeding and Husbandry: Morphological analysis was conducted “only on kidneys isolated from P1 and P15 mice” as described below.

Reviewer #2: This study presents interesting results by using murine models to demonstrate that Vangl2 gene knock-out (KO) in collecting duct causes tubular dilation and microcysts in kidney during embryogenic tubulogenesis due to defects in convergent extension and apical constriction. The tissue section microscopy data (immuno-fluorescence as well as IHC) and the corresponding quantified graphs showed structural defects in Vangl2 exon 4-excised mice at E17.5 and P1 stages, yet such defects appeared to get improved and barely lead to corresponding abnormality in kidney as the mice developed.

I find the conclusion drawn on Vangl2 gene solid and convincing based on extensive mouse models included in the study for comparison. LooptailS464N and Vangl2∆/∆ (exon 4 KO) mice were used for ubiquitous abrogation of Vangl2 function; both mice resulted in consistent phenotypes and with lethality at E18.5, hence the authors further used the Vangl2∆/CD (exon 4 conditionally KO in collecting ducts) model for studying the kidney defects. Proper control mice groups (Cre+;Vangl2+/+ and Cre-;Vangl2Fl/Fl) were included - although the mice being evaluated are of mixed background (Bl6/CD-1) to various degrees, the authors made efforts to backcross the Vangl2∆/CD to Bl6 strain to mitigate the potential effects. An additional model Vangl2∆TM/LoxP (one allele∆TM has ubiquitous exon 4 KO whereas the second allele LoxP has exon 2 and 3 conditionally KO in entire nephron; exon 2 contains the start codon) was included and consistent phenotypic outcomes were observed as those in Vangl2∆/CD group, further elucidating the role of Vangl2 gene in embryonic renal tubules.

Overall, this paper presented in-depth analysis and I believe that it is suitable to be published in PLOS ONE.

Following are minor issues in the manuscript that need to be amended:

1. Supplemental Figure Legend 2 mentioned ‘white arrow’: should be changed to ‘black arrow’.

2. 3rd paragraph in the introduction: acronym ‘UB’ appearing for the first time requires its expansion (i.e. ureteric bud).

6. PLOS authors have the option to publish the peer review history of their article (what does this mean?). If published, this will include your full peer review and any attached files.

Reviewer #1: No

Reviewer #2: Yes: Dorjee T.N. Shola

---

## [Author Response · Author response to Decision Letter 0]

31 Jan 2020

Response to reviewers has been appended to the manuscript file page

---

## [Editor Report · Decision Letter 1]

4 Mar 2020

Differential Role of Planar Cell Polarity Gene Vangl2 in Embryonic and Adult Mammalian Kidneys

PONE-D-19-31683R1

Dear Dr. Torban,

We are pleased to inform you that your revised manuscript has been judged scientifically suitable for publication and will be formally accepted for publication once it complies with all outstanding technical requirements.

With kind regards,

Shree Ram Singh, Ph. D.

Academic Editor

PLOS ONE
---

## [Editor Report · Acceptance letter]

9 Mar 2020

PONE-D-19-31683R1 

Differential Role of Planar Cell Polarity Gene Vangl2 in Embryonic and Adult Mammalian Kidneys 

Dear Dr. Torban:

I am pleased to inform you that your manuscript has been deemed suitable for publication in PLOS ONE. Congratulations! Your manuscript is now with our production department. 

With kind regards,

on behalf of

Dr. Shree Ram Singh 

Academic Editor

PLOS ONE